# T-Type Ca^2+^ Enhancer SAK3 Activates CaMKII and Proteasome Activities in Lewy Body Dementia Mice Model

**DOI:** 10.3390/ijms22126185

**Published:** 2021-06-08

**Authors:** Jing Xu, Ichiro Kawahata, Hisanao Izumi, Kohji Fukunaga

**Affiliations:** 1Departments of Pharmacology, Graduate School of Pharmaceutical Sciences, Tohoku University, Sendai 980-8578, Japan; xu.jing.q7@dc.tohoku.ac.jp (J.X.); kawahata@tohoku.ac.jp (I.K.); hisanao0413@gmail.com (H.I.); 2Departments of CNS Drug Innovation, Graduate School of Pharmaceutical Sciences, Tohoku University, Sendai 980-8578, Japan

**Keywords:** alpha-synuclein, Lewy body dementia, proteasome activity, Alzheimer’s disease, amyloid β plaque, SAK3, T-type Ca^2+^ channel enhancer

## Abstract

Lewy bodies are pathological characteristics of Lewy body dementia (LBD) and are composed of α-synuclein (α-Syn), which is mostly degraded via the ubiquitin–proteasome system. More importantly, 26S proteasomal activity decreases in the brain of LBD patients. We recently introduced a T-type calcium channel enhancer SAK3 (ethyl-8-methyl-2,4-dioxo-2-(piperidin-1-yl)- 2H-spiro[cyclopentane-1,3-imidazo [1,2-a]pyridin]-2-ene-3-carboxylate) for Alzheimer’s disease therapeutics. SAK3 enhanced the proteasome activity via CaMKII activation in amyloid precursor protein knock-in mice, promoting the degradation of amyloid-β plaques to improve cognition. At this point, we addressed whether SAK3 promotes the degradation of misfolded α-Syn and the aggregates in α-Syn preformed fibril (PFF)-injected mice. The mice were injected with α-Syn PFF in the dorsal striatum, and SAK3 (0.5 or 1.0 mg/kg) was administered orally for three months, either immediately or during the last month after injection. SAK3 significantly inhibited the accumulation of fibrilized phosphorylated-α-Syn in the substantia nigra. Accordingly, SAK3 significantly recovered mesencephalic dopamine neurons from cell death. Decreased α-Syn accumulation was closely associated with increased proteasome activity. Elevated CaMKII/Rpt-6 signaling possibly mediates the enhanced proteasome activity after SAK3 administration in the cortex and hippocampus. CaMKII/Rpt-6 activation also accounted for improved memory and cognition in α-Syn PFF-injected mice. These findings indicate that CaMKII/Rpt-6-dependent proteasomal activation by SAK3 recovers from α-Syn pathology in LBD.

## 1. Introduction

Lewy body dementia (LBD) is the second most common neurodegenerative disease worldwide. In addition to motor dysfunctions such as trembling and slow movements, non-motor dysfunctions, including dementia, depression, and anxiety, are also observed in LBD patients [1]. The main pathological features of LBD are α-synuclein (α-Syn) neuronal inclusions, such as the presence of Lewy bodies (LBs) and neuronal loss [2]; α-Syn is the principal constituent of LBs [3]. There are many forms of α-Syn, such as monomers, oligomers, fibers, and other conformations [4]. Previous reports have suggested that α-Syn oligomers exhibit toxicity in vitro and in vivo [5,6]. A study using a murine model showed that the injection of α-Syn preformed fibrils (PFFs) into the striatum spread to the substantia nigra (SN) [7], similar to prion proteins, leading to the loss of dopamine neurons. Neuronal death and loss of neuronal circuits in the striatum induce cognitive and motor impairments in mice [8]. Therefore, α-Syn plays a crucial role in the pathogenesis and progression of LBD and PD.

T-type voltage-gated calcium channels (T-VGCC), which play a critical role in brain function [9,10], have three electrophysiological characteristics: fast inactivation, slow deactivation kinetics, and low single-channel conductance [11,12]. Three types of T-type calcium channels consisting of Ca_v_3.1, Ca_v_3.2, and Ca_v_3.3 are encoded by *CACNA1G*, *CACNA1H*, and *CACNA1I* genes, respectively, which are expressed in the brain and play essential roles in both physiological and pathological systems, including sleep, pain, and epilepsy [13,14,15,16,17]. Our previous studies confirmed that SAK3 stimulates T-type voltage-gated calcium channels in Ca_v_3.1 and Ca_v_3.3 without affecting Ca_v_3.2 [18]. We also showed that SAK3 improved the impaired cognitive function in olfactory bulbectomized mice via the action of T-VGCC using murine Alzheimer’s disease (AD) models [18,19]. Recently, Izumi et al. reported that SAK3 administration ameliorated cognitive impairments in App NL-F/NL-F(NL-G-F) knock-in mice by improving the synaptic abnormalities in the hippocampal CA1 and cortex. Moreover, they found that SAK3 administration reduced Aβ deposition, which has deleterious effects on the brain, by reducing the proteasome activity [20,21]. The prominent observation in amyloid precursor protein knock-in (APP-KI) mice is that SAK3 administration enhances Ca^2+^/calmodulin-dependent protein kinase II (CaMKII)/Rpt6 phosphorylation, thereby rescuing the decreased proteasome activity in the APP-KI mouse brain.

The ubiquitin–proteasome system (UPS) is known as an efficient pathway for degrading misfolded or damaged proteins, and it plays a significant role in maintaining the physiological functions of cells [22,23]. UPS dysfunction is often observed in neurodegenerative diseases, such as AD [24] and PD [25,26], with protein misfolding and aggregation. Previous reports have shown that a frameshift mutation in the ubiquitin transcript was explicitly observed in the brains of patients with AD [27,28]. Aβ oligomers can inhibit proteasome activity by directly binding with the 20S proteasome [29]. Previous studies have shown that α-Syn aggregation can directly inhibit the proteolytic active site of the β subunit of the 20S proteasome [30] or bind it to the S6 subunit of the 19S cap [31]. It is uncertain whether protein aggregation leads to proteasomal damage, or vice versa. Under normal physiological conditions, intracellular α-Syn is mostly degraded by the UPS [32,33]. In addition, many pathological symptoms have been observed in PD, such as the formation of Lewy body-like inclusions in rats that received systemic administration of the proteasome inhibitor [34]. However, tissues from clinically diagnosed PD patients were checked and compared with normal aging controls; the expression level and activity of the 20S proteasome decreased significantly in PD nigral neurons that contain α-Syn inclusions [27], and the expression level and activity of the 19S Rpt6 subunit expression was also found in PD patients [35]. In several studies on brain aging, memory deficits with age are associated with reduced phosphorylation of the Rpt6 proteasome regulatory subunit [36]. However, no significant difference in baseline 20S proteasome activity was observed in the hippocampus of young and elderly rats [37].

CaMKII regulates Rpt6 phosphorylation and proteasome activity during the formation of long-term fear memory [38]. Moreover, CaMKII regulates memory formation in the amygdala and phosphorylation of Rpt6, a component of the proteasome 19S subunit [38,39]. Inhibition of CaMKII decreases the rate of new spine growth, and the mutation of Ser120 in the Rpt6 blocks the growth of new spines [38,39]. Considering the SAK3-mediated CaMKII activity, we also investigated the effects of SAK3 on UPS dysfunction in relation to PD.

## 2. Results

### 2.1. SAK3 Prevented the Development of Phosphorylated α-Syn (Ser129) of SNc in PF-Injected Mice

Immunohistochemistry was performed 12 weeks after PFF injection, as described in the schedule shown in Figure 1, and the spread of phosphorylated α-Syn in the substantia nigra pars compacta (SNc) area was analyzed. In the PBS group, phosphorylated α-syn was not detected in any area of the brain. The existing regions of phosphorylated α-Syn in the brain are similar to those reported previously [40]. Phosphorylated α-Syn in PFF-injected mice registered at high levels in the SNc area (3-month administration [3M]: 61.46 ± 6.578, *p <* 0.0001; 1-month administration [1M]: 73.24 ± 6.726, *p <* 0.0001; vs. PBS + vehicle; Figure 2A–C), and was decreased by SAK3 administration for 3 months (0.5 mg/kg: 31.24 ± 5.439, *p* = 0.0017; 1.0 mg/kg: 29.32 ± 4.611, *p* = 0.0004; vs. PFF + vehicle; Figure 2A,B) and 1 month (1.0 mg/kg: 23.58 ± 5.503, *p <* 0.0001; vs. PFF + vehicle; Figure 2A,C).

### 2.2. SAK3 Attenuated Neuronal Death and α-Syn Aggregation of Dopaminergic Neurons in PFF-Injected Mice

We checked the level of α-Syn aggregation in dopaminergic neurons. There was no α-Syn aggregation detected in any area of the brain of the PBS-injected mice. On the other hand, in PFF-injected mice, high levels of anti-phosphorylated α-Syn-positive aggregations were observed in the SNc area (3M: 74.97 ± 7.449, *p <* 0.0001; 1M: 72.81 ± 8.007, *p <* 0.0001; vs. PBS + vehicle; Figure 3A–C). More importantly, these phosphorylated α-Syn-positive aggregates were decreased by the administration of SAK3 for 3 months (0.5 mg/kg: 30.9 ± 4.414, *p <* 0.0001; 1.0 mg/kg: 9.67 ± 3.007, *p <* 0.0001; vs. PFF + vehicle; Figure 3A,B) and for 1 month (1.0 mg/kg: 25.28 ± 4.806, *p* = 0.0001; vs. PFF + vehicle; Figure 3A,C). These data indicate that SAK3 protects against the formation of phosphorylated α-Syn-positive aggregates in the midbrain.

### 2.3. SAK3 Prevented the Dopaminergic Neuronal Death of SNc in PFF-Injected Mice

The aggregation of accumulated α-Syn leads to the loss of TH protein [41]. Thus, we investigated the TH-positive SNc cells in the mouse brain and analyzed the protective effect of SAK3. As expected, the loss of dopaminergic neurons in the SNc area was observed (3M: 69.16 ± 3.66, *p <* 0.0001; 1M: 74.96 ± 6.902, *p <* 0.0001; vs. PBS + vehicle; Figure 4B,D,E,G). Intriguingly, SAK3 administration for 3 months prevented the loss of TH immunoreactivity (0.5 mg/kg, 105.7 ± 8.651, *p* = 0.0153; 1.0 mg/kg: 127.7 ± 7.327, *p <* 0.0001; vs. PFF + vehicle; Figure 4B,D), as well as after 1 month of SAK3 administration (1.0 mg/kg: 116.8 ± 6.397, *p* = 0.0147; vs. PFF + vehicle; Figure 4E,G), whereas no alteration in the ventral tegmental area (VTA) was observed (Figure 4C,F).

### 2.4. SAK3 Improved the Reduction of Proteasome Activity through the CaMKII-Rpt6 Signaling Pathway in PFF-Injected Mice

The degradation of α-Syn is mediated by the proteasome system [41]. To assess its therapeutic potential for α-Syn degradation, we examined the effect of SAK3 on proteasomal activity. In the proteasome activity assay, we found that proteasome activity was decreased in PFF-injected mice (Suc-LLVY-3M: 53.26 ± 7.697, *p* = 0.0186; Suc-LLVY-1M: 66.72 ± 4.866, *p* = 0.0116; Bz-VGR-3M: 39.79 ± 6.874, *p* = 0.042; Bz-VGR-1M: 65.4 ± 7.47, *p* = 0.0267; Z-LLE-3M: 35.22 ± 8.944, *p* = 0.004; Z-LLE-1M: 48.84 ± 3.258, *p* = 0.0243; vs. PBS + vehicle; Figure 5A–F), and was improved significantly by SAK3 administration (Suc-LLVY-3M at 0.5 mg/kg: 102.7 ± 13.72, *p* = 0.0211; Suc-LLVY-3M at 1.0 mg/kg: 115.2 ± 9.434, *p* = 0.0026; Suc-LLVY-1M at 1.0 mg/kg: 97.38 ± 10.42, *p* = 0.0211; Bz-VGR-3M at 0.5 mg/kg: 97.52 ± 16.05, *p* = 0.0089; Bz-VGR-3M at 1.0 mg/kg: 91.57 ± 15.82, *p* = 0.0223; Bz-VGR-1M at 1.0 mg/kg: 101.9 ± 11.3, *p* = 0.0183; Z-LLE-3M at 0.5 mg/kg: 95.87 ± 11.78, *p* = 0.0106; Z-LLE-3M at 1.0 mg/kg: 110 ± 18.35, *p* = 0.0013; Z-LLE-1M at 1.0 mg/kg: 102.6 ± 14.5, *p* = 0.0169; vs. PFF + vehicle; Figure 5A–F).

CaMKII-mediated phosphorylation of Rpt6 plays a key role in long-term memory formation and new spine growth [38,39]. Considering that CaMKII autophosphorylation was enhanced by SAK3, we investigated the activity of the proteasome. First, we found that there was no difference between the total CaMKII and Rpt6 levels in the cortex (data not shown). Moreover, the results also showed that the autophosphorylation of CaMKII and phosphorylation of Rpt6 (S120) decreased in PFF-injected mice (p-CaMKII/CaMKII-3M: 50.03 ± 2.548, *p* = 0.0002; p-CaMKII/CaMKII-1M: 53.6 ± 7.934, *p* = 0.0001; p-Rpt6/Rpt6-3M: 53.91 ± 2.7, *p* = 0.0031; p-Rpt6/Rpt6-1M: 55.08 ± 8.052, *p* = 0.0019; vs. PBS + vehicle; Figure 6A–F), and were improved significantly by SAK3 administration (p-CaMKII/ CaMKII-3M at 0.5 mg/kg: 79.5 ± 5.575, *p* = 0.0398; p-CaMKII/CaMKII-3M at 1.0 mg/kg: 92.99 ± 8.323, *p* = 0.0023; p-CaMKII/CaMKII-1M at 1.0 mg/kg: 87.85 ± 3.232, *p* = 0.002; p-Rpt6/Rpt6-3M at 0.5 mg/kg: 97.14 ± 6.781, *p* = 0.0053; p-Rpt6/Rpt6-3M at 1.0 mg/kg: 88.85 ± 1.507, *p* = 0.0259; p-Rpt6/Rpt6-1M at 1.0 mg/kg: 100.3 ± 8.212, *p* = 0.0017; vs. PFF + vehicle; Figure 6A–F).

### 2.5. SAK3 Ameliorated Motor and Cognitive Impairments in PFF-Injected Mice

Behavioral tests were conducted following the 3-month SAK3 treatment schedule (Figure 1C) after PFF injection. Motor and cognitive functions were investigated at 4, 8, and 12 weeks after PBS or α-Syn injection.

In the rotarod test, the latency time of stopping the rotarod was shortened in α-Syn PFF-injected mice (8 weeks: 118.7 ± 24.08, *p* < 0.0001; 12 weeks: 116.5 ± 16.6, *p* < 0.0001; vs. PBS + vehicle; Figure 7A) and recovered by SAK3 administration (0.1 mg/kg-8 weeks: 222.1 ± 24.79, *p* = 0.0095; 0.1 mg/kg-12 weeks: 215 ± 25.67, *p* = 0.0007; 0.5 mg/kg-8 weeks: 270.3 ± 19.21, *p* < 0.0001; 0.5 mg/kg-12 weeks: 278.8 ± 15.57, *p* < 0.0001; 1.0 mg/kg-8 weeks: 262.8 ± 17.51, *p* < 0.0001; 1.0 mg/kg-12 weeks: 287.1 ± 9.904, *p* < 0.0001; vs. PFF + vehicle; Figure 7A). Likewise, in the beam-walking test, the frequency of foot slips increased in α-Syn PFF-injected mice (4 weeks: 1.4 ± 0.2667, *p* = 0.0436; 8 weeks: 2.3 ± 0.3667, *p* = 0.0059; 12 weeks: 2.7 ± 0.423, *p* = 0.0003; vs. PBS + vehicle; Figure 7B) and was recovered by SAK3 administration (0.1 mg/kg-8 weeks: 1.1 ± 0.2769, *p* = 0.0138; 0.1 mg/kg-12 weeks: 1.1 ± 0.2769, *p* = 0.0008; 0.5 mg/kg-8 weeks: 0.7692 ± 0.2011, *p* = 0.0003; 0.5 mg/kg-12 weeks: 0.7692 ± 0.2011, *p <* 0.0001; 1.0 mg/kg-4 weeks: 0.4615 ± 0.1439, *p* = 0.0181; 1.0 mg/kg-8 weeks: 0.5385 ± 0.1439, *p <* 0.0001; 1.0 mg/kg-12 weeks: 0.5385 ± 0.1439, *p <* 0.0001; vs. PFF + vehicle; Figure 7B).

Furthermore, the memory function of α-Syn PFF-injected mice was also checked. In the novel object recognition task, no groups showed differences in the discrimination index for the training session (data not shown). However, the discrimination index in PFF-injected mice decreased (4 weeks: 49.39 ± 1.157, *p* = 0.0036; 8 weeks: 49.16 ± 0.7883, *p* = 0.0005; 12 weeks: 49.85 ± 0.652, *p* = 0.015; vs. PBS + vehicle; Figure 7C), and was reduced by SAK3 administration in the test session (0.5 mg/kg-8 weeks: 59.52 ± 2.746, *p* = 0.0117; 0.5 mg/kg-12 weeks: 60.12 ± 2.331, *p* = 0.0143; 1.0 mg/kg-4 weeks: 56.6 ± 0.9874, *p* = 0.0227; 1.0 mg/kg-8 weeks: 59.72 ± 1.834, *p* = 0.0061; 1.0 mg/kg-12 weeks: 59.44 ± 2.635, *p* = 0.018; vs. PFF + vehicle; Figure 7C). In the passive avoidance task, there was no change in the latency time to enter the dark compartment among all groups (data not shown) during the training session. In the test session, the latency time of entering the dark compartment was shortened after electrical stimulation (48.5 ± 16.22, *p <* 0.0001; vs. PBS + vehicle; Figure 7D) and recovered by SAK3 administration (0.5 mg/kg: 176.8 ± 28.58, *p* = 0.0115; 1.0 mg/kg: 218.3 ± 16.52, *p* = 0.0002; vs. PFF + vehicle; Figure 7D). Interestingly, in the Y-maze task, there was no change in the percentage of alternation behaviors among all groups. The times of total arm entries increased at 12 weeks after PFF injection (23.7 ± 2.098, *p* = 0.0493; vs. PBS + vehicle) and was recovered by SAK3 (1.0 mg/kg) administration (17.31 ± 1.278, *p* = 0.0306; vs. PFF + vehicle; data not shown).

For the 1-month SAK3 treatment after PFF injection (Figure 1D), behavioral tests were also performed (Figure 8). In the rotarod test, the latency time of stopping the rotarod was shortened in α-Syn PFF-injected mice (4 weeks: 130.4 ± 34.31, *p* = 0.0452; 102.7 ± 23.71, *p* = 0.0161; 128.4 ± 33.42, *p* = 0.0404; 8 weeks: 109.4 ± 18.81, *p* = 0.0011; 128.7 ± 41.1, *p* = 0.0095; 121.4 ± 29.06, *p* = 0.0028; 129.3 ± 31.57, *p* = 0.0069; 12 weeks: 78.56 ± 15.52, *p <* 0.0001 vs. PBS + vehicle; Figure 8A) and was recovered by SAK3 (1.0 mg/kg) administration (257.4 ± 27.36, *p <* 0.0001; vs. PFF + vehicle; Figure 8A). In the beam-walking test, the number of foot slips increased in α-Syn PFF-injected mice (8 weeks: 2.222 ± 0.4006, *p* = 0.0187; 0.0233 ± 0.3595, *p* = 0.0233; 2.111 ± 0.3093, *p* = 0.0336; 2.375 ± 0.4605, *p* = 0.0107; 12 weeks: 3.444 ± 0.3379, *p <* 0.0001; vs. PBS + vehicle; Figure 8B) and was recovered by SAK3 (1.0 mg/kg) administration (1.125 ± 0.2266, *p <* 0.0001; vs. PFF + vehicle; Figure 8B). In the novel object recognition task, no group showed differences in the discrimination index for the training session (data not shown). However, the discrimination index in PFF-injected mice decreased (4 weeks: 47.95 ± 1.077, *p* = 0.0137; 46.65 ± 3.468, *p* = 0.0088; 48.05 ± 2.433, *p* = 0.0192; 8 weeks: 47.08 ± 2.506, *p* = 0.0002; 49.34 ± 2.903, *p* = 0.0044; 52.31 ± 1.089, *p* = 0.0443; 50.17 ± 1.455, *p* = 0.0071; 12 weeks: 42.26 ± 2.739, *p* = 0.0025; vs. PBS + vehicle; Figure 8C), and was reduced by SAK3 (1.0 mg/kg) administration in the test session (62.57 ± 1.132, *p* = 0.0466; vs. PFF + vehicle; Figure 8C). In the passive avoidance tasks, there were no changes in the latency times to enter the dark compartment among all groups during the training session (data not shown). However, in the test session, the latency times to enter the dark compartment were shortened after electrical stimulation (69.78 ± 28.62, *p* = 0.0037; vs. PBS + vehicle; Figure 8D) and recovered by SAK3 (1.0 mg/kg) administration (195.8 ± 19.37, *p* = 0.0174; vs. PFF + vehicle; Figure 8D).

## 3. Discussion

In this study, because oral SAK3 administration inhibits amyloid β plaque formation in APP-KI mice by activating the proteasome activity [20,21], we investigated whether SAK3 promotes the degradation of fibril α-Syn in PFF-injected LBD model mice. If SAK3 administration promotes the degradation of misfolded proteins, the therapeutics have the potential to solve the problems of diverse protein misfolding diseases such as PD, LBD, tauopathies, and Huntington diseases in addition to AD. As expected, chronic SAK3 administration rescued the impaired CaMKII-Rpt6 signaling in LBD model mice after injection of PFF. However, even after the onset of cognitive impairment, SAK3 administration significantly prevented the progression of LBD behaviors in both motor dysfunction and cognition.

Although the mechanism of α-Syn toxicity in LBD is controversial, α-Syn oligomers can exhibit toxicity [5,6,42] via mitochondrial dysfunction by altering cell membrane permeability [43], or by inducing lysosomal leakage [44]. There are several ways to convert monomers of α-Syn to oligomers, such as the phosphorylation (Ser129) [45] and oxidative modification of α-Syn [46]. For example, less than 4% of α-Syn are phosphorylated at Ser129 in healthy human brains, while more than 90% of α-Syn are phosphorylated in the brains of PD patients [47,48]. The formation of insoluble filaments was checked in vitro, and the phosphorylation of α-Syn at Ser129 was found to have a causative function in fibrillization compared to non-phosphorylated α-Syn [47]. In a *Drosophila* model study, mutating Ser129 revealed a complete inhibition of inclusion body formation and neuronal death following human α-Syn expression [49]. The phosphorylation levels of α-Syn (Ser129) are closely associated with the generation of LBs and neurodegenerative changes in dopamine neurons in the present study. SAK3 administration reduced the levels of phosphorylated α-Syn in dopaminergic neurons in the SNc. Although SAK3 directly mediated the inhibition of α-Syn phosphorylation, the degradation of α-Syn fibrils primarily resulted in reduced phosphorylation.

The 26S proteasome is a cylindrical complex that consists of a 20S core particle and a 19S regulatory particle [50], composed of α and β subunits [51]. In PD, it has been shown that the expression of mutations or wild α-Syn, particularly in the conformation of soluble oligomers and aggregates, inhibits the activity of 20S or 26S proteasomes [52]. The report also showed that the function of UPS and 19S proteasome ATPase Rpt6 decreased in 1-methyl-4-phenyl-1,2,3,6-tetrahydropyridine (MPTP)-treated PD model mice [53,54]. Although the mechanism remains unclear, the authors speculate that mitochondrial dysfunction induced by MPTP affects the UPS function, a system responsible for the selective proteolytic degradation of misfolded proteins from various intracellular compartments, including mitochondria [55]. The α-Syn oligomers have the potential to unfold through the open-gated channel of the 26S proteasome and insert themselves into the catalytic chamber, directly inhibiting the proteolytic active site of the β subunit of the 20S proteasome [30].

Likewise, aggregated α-Syn selectively binds with the proteasomal protein S6, a subunit of the 19S cap [31]. Reduced Rpt6 subunit expression was found in three brain regions of LBD, PD, and AD patients and was associated with reduced proteasome activity [35]. However, the reason for the reduced expression of the Rpt6 subunit is not clear. More importantly, strong associations were observed between Rpt6 levels and cognitive impairment [35]. We observed that Rpt6, a component of the proteasome 19S subunit, is phosphorylated by CaMKII (Figure 6) in the brains of PFF-injected mice. Therefore, proteasome activity, which is regulated by the CaMKII-Rpt6 pathway, may be a new target for neurodegenerative disease therapy.

We previously discovered the presence of CaMKII in rat brain [56,57], and its activity strengthened the function of neuronal networks in memory by inducing hippocampal long-term potentiation [58]. We further developed a novel T-VGCC enhancer, SAK3, which antagonizes the reduction of CaMKII phosphorylation levels via T-VGCC enhancement, and improved memory deficits in olfactory bulbectomized mice [19]. Moreover, SAK3 improved memory and cognitive function by increasing the autophosphorylation of CaMKII and restoring spine abnormalities in APP^NL-G-F^ knock-in mice [20]. Moreover, SAK3 inhibited amyloid-beta (Aβ) accumulation and aggregation in APP^NL-G-F^ knock-in mice [59]. The improved proteasome activity was mediated by the CaMKII/Rpt6 signaling pathway in APP transgenic mice [20].

This study successfully demonstrated that SAK3 reduced the levels of aggregated α-Syn in PFF-injected mice and increased proteasome activity through Rpt6 phosphorylation. We hypothesized that the T-type calcium enhancer SAK3 may work in PD, as shown in Figure 9. SAK3 triggers intracellular calcium influx and promotes Glu release by enhancing T-type calcium channels and increases proteasome activity via the CaMKII/Rpt6 signaling pathway. Moreover, proteasome activity contributes to the aggregation of α-Syn degradation and spine improvement. In conclusion, proteasome activity is promoted by SAK3 by mediating CaMKII activity. Because the detailed mechanism underlying the CaMKII-mediated degradation of aggregated α-Syn is still unclear, we will solve these problems in future studies. PD therapies focus on improving motor function. For instance, levodopa can ameliorate PD symptoms by increasing dopamine levels [60]. However, levodopa did not improve the cognitive function in MPTP-treated PD model mice [61]. Therefore, a new kind of drug such as SAK3, which can degrade the aggregated α-Syn by activating proteasome activity, will greatly impact PD and LBD therapies.

## 4. Materials and Methods

### 4.1. Animals and Murine Model

Adult male C57BL/6J mice (8–12 weeks old) were obtained from Clea Japan, Inc. (Tokyo, Japan) and housed under conditions of constant temperature (23 ± 2 °C) and humidity (55 ± 5%) on 12 h light and dark cycles (lights on 9:00 a.m.–9:00 p.m.). Animals were provided with food and water ad libitum. All animal studies were conducted in accordance with the Institutional Animal Care and Use Committee of the Tohoku University Environmental and Safety Committee [2019PhLM0-021 (approval date: 1 December 2019) and 2019PhA-024 (approval date: 1 April 2019)]. The mice were operated as described previously [18]. In brief, α-Syn PFF (5 µg/each area) was injected into the bilateral striatum (anterior, 0.7 mm; lateral, ± 1.9 mm; depth −2.5 mm [62] relative to the bregma). Mice were injected with PBS as a control group for comparison.

### 4.2. Preformed Fibrillization of α-Syn

PFF α-Syn was prepared as previously described [63,64]. Briefly, mouse monomers were dissolved in PBS and centrifuged for 1 h at 100,000× *g* at 4 °C. Then, the supernatants were shaken at 200 repetitions/min for 7 days. The fibrilized α-Syn was sonicated with 10% power for 30 s using an ultrasonic homogenizer (SONIFIER Model 250 Advanced: Branson, Danbury, CT, USA) [18,65] and stored at −80 °C until use.

### 4.3. Drug Administration and Experimental Design

SAK3 was dissolved in distilled water. As shown in Figure 1, animals were administered SAK3 orally once daily at 0.1, 0.5, or 1.0 mg/kg of the drug at 10 mL/kg, or the same volume of distilled water (for controls), for 1 or 3 months. Mice were subjected to behavioral tests at 4, 8, and 12 weeks after PFF injection to assess motor function (including rotarod task and beam-walking task) and cognitive function (including Y-maze task, novel object recognition task, and step-through passive avoidance task).

### 4.4. Behavioral Analyses

#### 4.4.1. Rotarod Task

The rotarod task was performed as described previously [62]. The rotarod apparatus consisted of a base platform and a non-slippery rod (diameter, 3 cm; length, 30 cm). Before PFF injection, trained mice were placed on a rod rotating at 20 rpm, and the process was repeated until the fall latency exceeded 100 s. In the test session, the falling latency was measured for 300 s.

#### 4.4.2. Beam-Walking Task

The beam-walking task was performed according to a previously described method [63]. The apparatus consisted of a rectangular beam (length, 870 mm; width, 5 mm) and a goal box (155 mm × 160 mm × 5 mm). Before PFF injection, trained mice were placed 10 cm away from the goal and allowed to reach the goal box. The number of foot slips (missteps) from the end of the beam to the goal box was recorded.

#### 4.4.3. Y-Maze Task

Short-term spatial memory was investigated using the Y-maze task. As previously described [18,63], a mouse was placed at the end of the arm of a Y-maze and explored freely for 8 min. After each session, the objects and box were cleaned with 70% ethanol to prevent odor recognition. Alternations were defined as entries into all three arms on consecutive choices. The maximum number of alternations was defined as the total number of arms. The percentage of alternations was calculated as the actual alternations/maximum alternations ×100.

#### 4.4.4. Novel Object Recognition Task

Cognitive function was evaluated using the novel object recognition task according to a previously described method [18,62]. In the trial session, the mice were exposed to two similar objects (consisting of a wooden block) placed symmetrically at the center of the open field box for 10 min. After 1 h, one object was replaced by a novel object, and exploratory behavior was monitored again for 5 min during the test session. After each session, the objects and box were cleaned with 70% ethanol to prevent odor recognition. The timing of object exploration was defined by behaviors such as rearing on, touching, and sniffing. The discrimination index was analyzed using the ratio of exploratory contacts to familiar and novel objects.

#### 4.4.5. Step-through Passive Avoidance Task

The step-through passive avoidance task was conducted as described previously [18,63]. Briefly, after a mouse moved from the light to the dark compartment of a box, it received an electric shock (0.5 mA for 2 s) when passing through the floor, completing a trial session. To evaluate retention levels, mice were placed in the light compartment, and step-through latency times were recorded over 300 s after 24 h. To avoid stress effects, the step-through passive avoidance task was performed only 12 weeks after α-Syn PFF injection.

### 4.5. Immunohistochemistry

Immunohistochemistry was performed as described previously [63]. Mice were anesthetized, perfused with PBS, and fixed with 4% paraformaldehyde. After 24 h of post-fixation at 4 °C, brains were sectioned (50 μm thick) using a vibratome (Dosaka EM Co. Ltd., Kyoto, Japan). Brain slices were incubated with 0.01% Triton X-100 in PBS (pH 7.4) three times for 30 min and then washed with PBS.

Next, brain slices were incubated for 1 h with a blocking buffer (1% bovine serum albumin and 0.3% Triton-X in PBS), and treated with the primary antibody in blocking buffer for 3 days at 4 °C under shaking conditions. Antibodies included rabbit monoclonal anti-phosphorylated α-Syn (Ser129) (1:200, Abcam, Cambridge, UK), mouse monoclonal anti-TH (1:1000; Immunostar, Hudson, WI, USA), and rabbit monoclonal anti-α-Syn aggregate (1:2000, Abcam, Cambridge, UK). The brain slices were washed again with PBS and incubated with secondary antibodies, including Alexa 488 anti-rabbit IgG and Alexa 594 anti-mouse IgG (1:500 in blocking solution; Invitrogen, Waltham, MA, USA), under dark conditions overnight at 4 °C. After washing, brain slices were mounted on Vectashield (Vector Laboratories, Inc., Burlingame, CA, USA). Immunofluorescence images were analyzed using a confocal laser scanning microscope (Leica TCS SP8, Leica Microsystems, Wetzlar, Germany). To check reproducibility, two slices from each mouse were used for counting positive cells in the SNc and VTA areas (3.0–3.5 from bregma).

### 4.6. Western Blotting Analysis

Western blot analysis was performed as described previously [20,65]. Mouse brain tissues were dissected, frozen in liquid nitrogen, and stored at −80 °C until use. Brain samples were homogenized in ice-cold buffer containing 500 mM NaCl, 50 mM Tris-HCl (pH 7.5), 0.5% Triton X-100, 4 mM EGTA, 10 mM EDTA, 1 mM Na_3_VO_4_, 40 mM Na_2_P_2_O_7_·10H_2_O, 50 mM NaF, 100 nM calyculin A, 50 μg/mL leupeptin, 25 μg/mL pepstatin A, 50 μg/mL trypsin inhibitor, and 1 mM DTT. Samples were centrifuged at 15,000 rpm for 10 min at 4 °C, and the supernatant protein concentrations were determined using the Bradford assay. Samples were boiled for 3 min at 100 °C with Laemmli sample buffer (0.38 M Tris-HCl, pH 6.8, 30% 2-mercaptoethanol, 15% glycerol, 12% SDS, and 0.05% bromophenol blue). Equivalent amounts of proteins were loaded onto SDS-polyacrylamide gels and transferred to Immobilon polyvinylidene difluoride membranes (Merck Millipore Ltd., Darmstadt, Germany). After blocking with 5% fat-free milk in TTBS solution (50 mM Tris-HCl, pH 7.5, 150 mM NaCl) for 30 min at room temperature, membranes were incubated with anti-phospho-CaMKII (1:5000) [66], anti-CaMKII (1:5000) [66], anti-phospho-Rpt6 (1:500; MBS9429032, MyBioSource, San Diego, CA, USA), anti-Rpt6 (1:1000; BML-PW9265, Enzo Life Science, Farmingdale, NY, USA) in TTBS solution, and anti-β-actin (1:5000; A5551, Sigma-Aldrich, St. Louis, MO, USA) overnight at 4 °C. After washing, the membranes were incubated with the secondary antibodies in TTBS. Blots were developed using an ECL detection system, and protein signals were analyzed using Image Gauge software (version 3.41; Fuji Film, Tokyo, Japan).

### 4.7. Proteasome Activity Assay

Proteasome activity assays were performed as previously described [20]. Frozen brain tissues were homogenized in ice-cold buffer containing 20 mM Tris-HCl (pH 7.5), 5 mM EDTA, 500 mM NaCl, 5 mM MgCl_2_, 1% Triton X-100, 1 mM DTT, and 2 mM ATP, and then centrifuged at 15,000 rpm for 15 min at 4 °C. The supernatant protein concentrations were determined using the Bradford assay. Next, the protein was mixed with 200 μM fluorogenic peptides, Suc-LLVY-AMC (Millipore, Bedford, MA, USA) or Bz-VGR-AMC (Enzo Life Science) or Z-LLE-AMC (Enzo Life Science), and assay buffer (25 mM HEPES, pH 7.5, 0.5 mM EDTA, 0.05% NP-40), followed by incubation for 1 h at 37 °C [20,66]. Fluorescence of the samples was measured at 380/460 nm (Ex/Em) using a fluorometer (FlexStation 3 Multi-Mode Microplate Reader; Molecular Devices, San Jose, CA, USA).

### 4.8. Statistical Analysis

All data are presented as mean ± standard error of the mean (SEM). Comparisons among multiple groups were evaluated by one-way analysis of variance (ANOVA) followed by Tukey’s post hoc test using GraphPad Prism 7 (GraphPad Software, Inc., La Jolla, CA, USA).

## Figures and Tables

**Figure 1 ijms-22-06185-f001:**
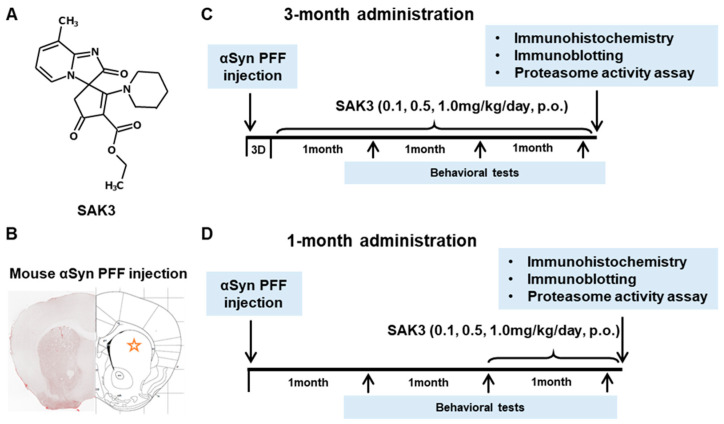
**Chemical structure of SAK3, experimental schedule, and α-Synuclein PFF injection area.** (**A**) Chemical structure of SAK3. (**B**) Position of PFF injection in the mouse brain striatum. Experimental schedule of (**C**) 3-month chronic administration or (**D**) 1-month administration of SAK3 in this study.

**Figure 2 ijms-22-06185-f002:**
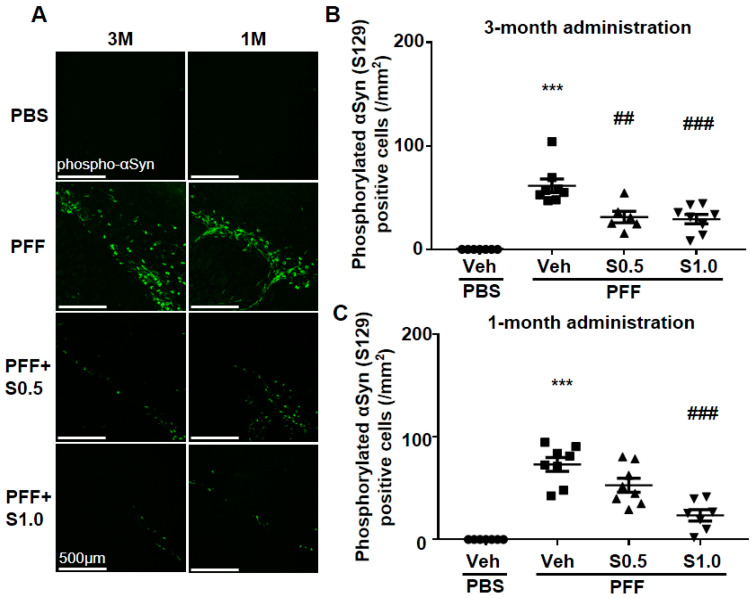
**SAK3 chronic administration prevents the spread of phosphorylated α-Syn in PFF-injected mice.** (**A**) Representative immunofluorescence images of phosphorylated α-Syn in the SNc region in both schedules of this study. Scale bar: 500 μm. The number of phosphorylated α-Syn-positive cells was counted in the SNc region for (**B**) 3 months of the SAK3 treatment schedule (*n* = 6–8 per group), and (**C**) 1 month of the SAK3 treatment schedule (*n* = 7–8 per group). Error bars represent SEM. *** *p* < 0.001 vs. vehicle-treated PBS injection mice; ^##^
*p* < 0.01, ^###^
*p* < 0.001 vs. vehicle-treated α-Syn PFF-injected mice. Abbreviations: Veh = vehicle; S = SAK3.

**Figure 3 ijms-22-06185-f003:**
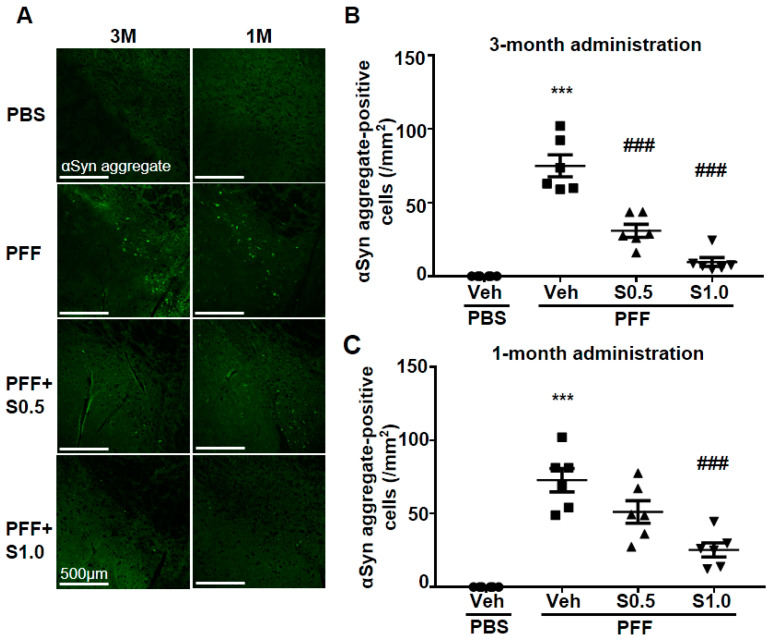
**SAK3 treatment prevents the spread of aggregated α-Syn in α-Syn PFF-injected mice.** (**A**) Representative immunofluorescence images of fibrous α-Syn in the SNc region in both schedules of this study. Scale bar: 500 μm. The number of fibrous α-Syn-positive cells was counted in the SNc region for (**B**) 3 months of the SAK3 treatment schedule (*n* = 6–8 per group) and (**C**) 1 month of the SAK3 treatment schedule (*n* = 7–8 per group). Error bars represent SEM. *** *p* < 0.001 vs. vehicle-treated PBS-injected mice; ^###^
*p* < 0.01 vs. vehicle-treated α-Syn PFF-injected mice. Abbreviations: Veh = vehicle; S = SAK3.

**Figure 4 ijms-22-06185-f004:**
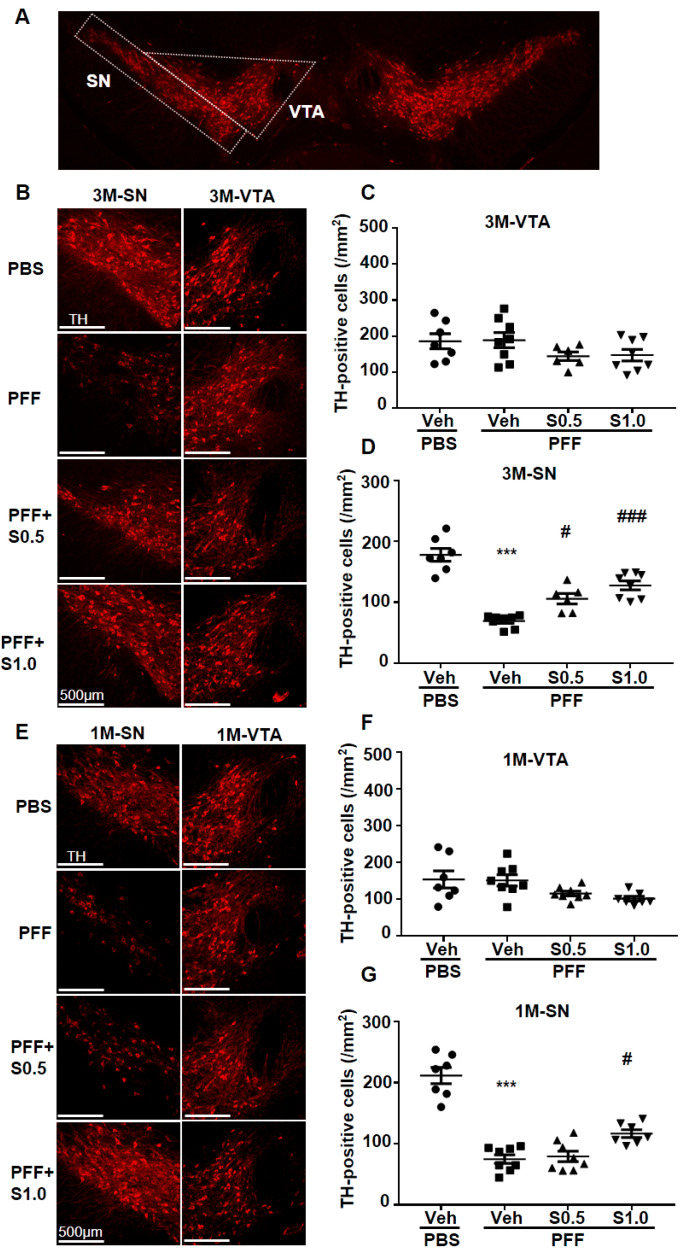
**SAK3 treatment prevents the dopaminergic neuronal death in the SN but not in the VTA of PFF-injected mice.** (**A**) The VTA and SNc region are framed by a white line. (**B**) The representative immunofluorescence images of TH in the VTA and SNc regions during the 3-month SAK3 treatment schedule. Scale bar: 500 μm. The number of TH-positive cells was counted in the (**C**) VTA and (**D**) SNc region (*n* = 6–8 per group). (**E**) The representative immunofluorescence images of TH in the VTA and SNc region during the 1-month SAK3 treatment schedule. Scale bar: 500 μm. The number of TH-positive cells was counted in the (**F**) VTA and (**G**) SNc region (*n* = 7–8 per group). Error bars represent SEM. *** *p* < 0.001 vs. vehicle-treated PBS-injected mice; ^#^
*p* < 0.05, ^###^
*p* < 0.001 vs. vehicle-treated α-Syn PFF-injected mice. Abbreviations: Veh = vehicle; S = SAK3.

**Figure 5 ijms-22-06185-f005:**
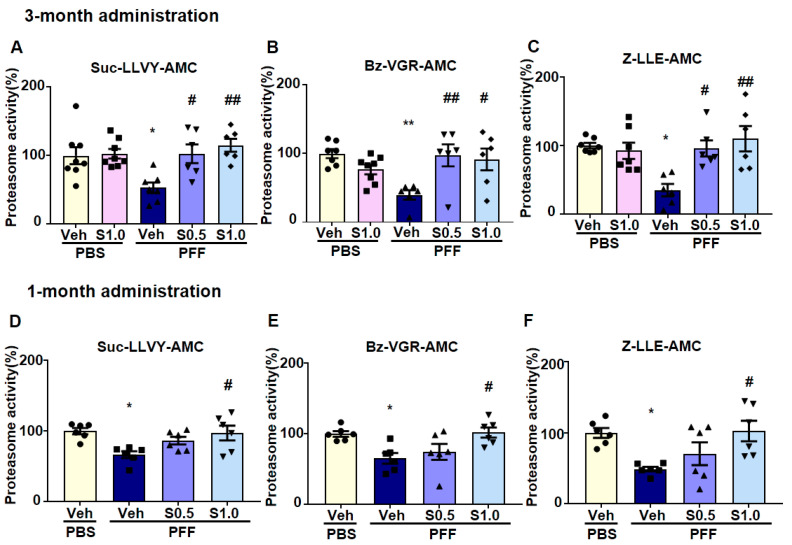
**SAK3 administration rescues the decrease in proteasome activity in α-Syn PFF-injected mice.** Proteasome activity assay using fluorogenic peptides (**A**,**D**) Suc-LLVY-AMC (chymotrypsin-like), (**B**,**E**) Bz-VGR-AMC (trypsin-like), and (**C**,**F**) Z-LLE-AMC (caspase-like) on the brain’s cortical region in both schedules (*n* = 6–8 per group). Error bars represent SEM. * *p* < 0.05, ** *p* < 0.01 vs. vehicle-treated PBS-injected mice; ^#^
*p* < 0.05, ^##^
*p* < 0.01 vs. vehicle-treated α-Syn PFF-injected mice. Abbreviations: Veh = vehicle; S = SAK3.

**Figure 6 ijms-22-06185-f006:**
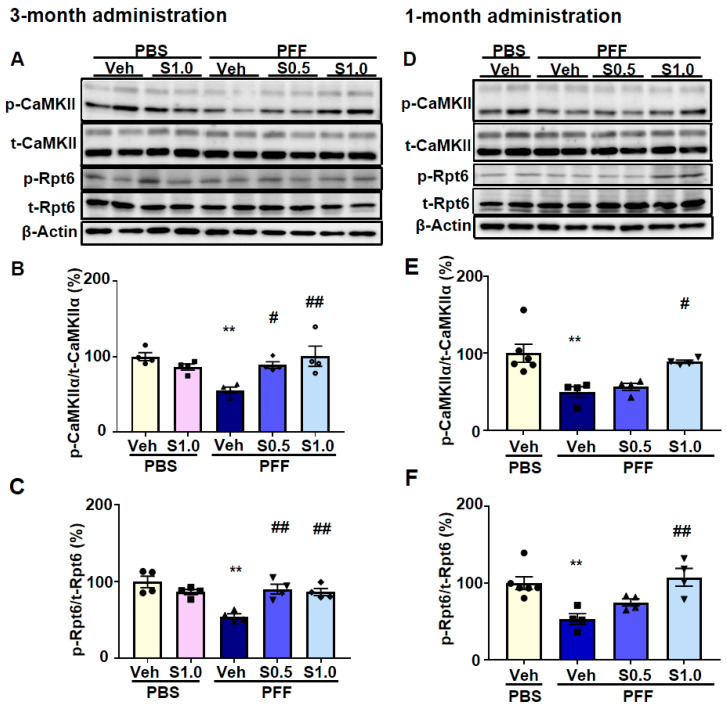
**SAK3 administration improves CaMKII-Rpt6 signaling in α-Syn PFF-injected mice.** (**A**,**D**) Representative images of western blot membranes containing cortical protein, probed with antibodies against autophosphorylated CaMKII (T286), CaMKII, phosphorylated Rpt6 (S120), Rpt6, and β-actin. Quantitative analyses of (**B**,**E**) autophosphorylated CaMKII (T286) and (**C**,**F**) phosphorylated Rpt6 (S120), during the 3-month SAK3 treatment schedule (*n* = 4 per group) and the 1-month SAK3 treatment schedule (*n* = 4–6 per group). Error bars represent SEM. ** *p* < 0.01 vs. vehicle-treated PBS-injected mice; ^#^
*p* < 0.05, ^##^
*p* < 0.01 vs. vehicle-treated α-Syn PFF-injected mice. Abbreviations: Veh = vehicle; S = SAK3.

**Figure 7 ijms-22-06185-f007:**
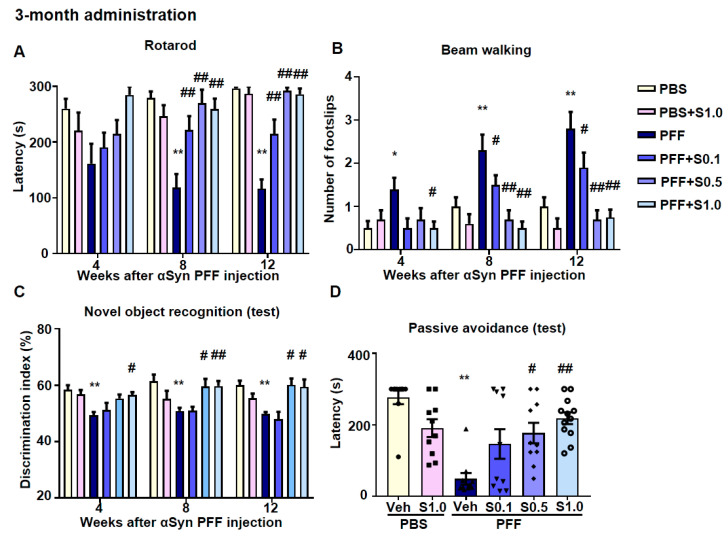
**Three months of SAK3 treatment improves cognitive and motor functions in α-Syn PFF-injected mice.** Analyses of motor function based on the (**A**) rotarod task (*n* = 10–12 per group) and (**B**) beam-walking task (*n* = 10–12 per group) at 4, 8, and 12 weeks after mouse α-Syn PFF injection. (**C**) Test session of the novel object recognition task (*n* = 10–12 per group). (**D**) Test session of the step-through passive avoidance task (*n* = 10–12 per group). Error bars represent SEM. * *p* < 0.05, ** *p* < 0.01 vs. vehicle-treated PBS-injected mice; ^#^
*p* < 0.05, ^##^
*p* < 0.01 vs. vehicle-treated α-Syn PFF-injected mice. Abbreviations: Veh = vehicle; S = SAK3.

**Figure 8 ijms-22-06185-f008:**
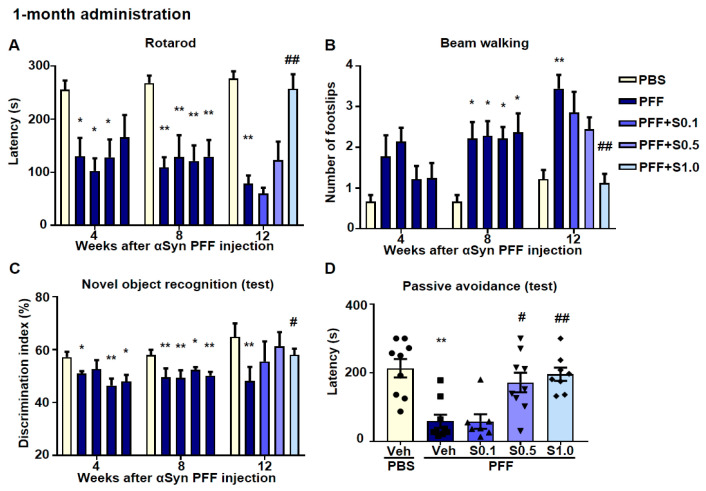
**High-concentration treatment with SAK3 for 1 month improves cognitive and motor functions in α-Syn PFF-injected mice.** Analyses of motor function based on the (**A**) rotarod task (*n* = 10–12 per group) and (**B**) beam-walking task (*n* = 10–12 per group) at 4, 8, and 12 weeks after mouse α-Syn PFF injection. (**C**) Test session of the novel object recognition task (*n* = 10–12 per group). (**D**) Test session of the step-through passive avoidance task (*n* = 10–12 per group). Error bars represent SEM. ^*^
*p* < 0.05, ^**^
*p* < 0.01 vs. vehicle-treated PBS injection mice; ^#^
*p* < 0.05, ^##^
*p* < 0.01 vs. vehicle-treated α-Syn PFF-injected mice. Abbreviations: Veh = vehicle; S = SAK3.

**Figure 9 ijms-22-06185-f009:**
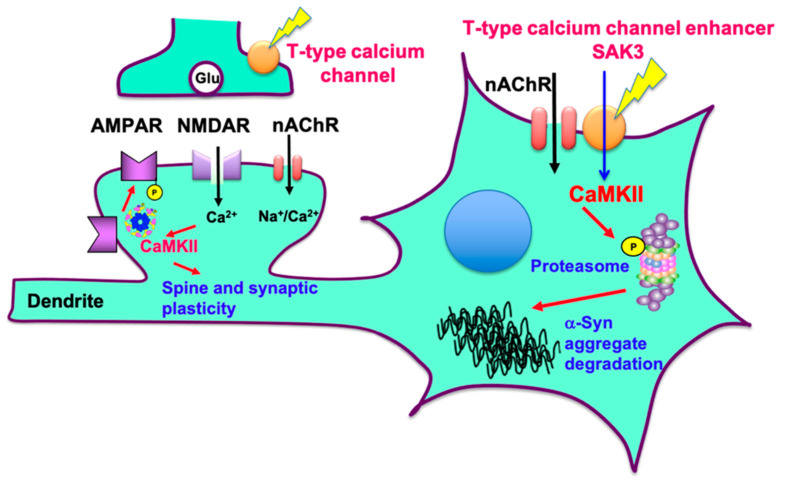
**Schematic illustration of the action mechanism of SAK3 in PD pathology.** SAK3 triggers intracellular calcium influx and promotes Glu release by enhancing T-type calcium channels. Proteasome activity is increased via the CaMKII/Rpt6 signaling pathway. Facilitated proteasomal activity by SAK3 contributes to the degradation of α-Syn aggregates and potentiates synaptic plasticity. AMPAR, glutamate α-amino-4-hydroxy-5-methyl-4-isoxazolepropionic acid receptor; NMDAR, glutamate N-methyl-D-aspartate receptor; nAChR, nicotinic acetylcholine receptor.

## Data Availability

The data presented in this study are available on request from the corresponding author.

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
