# Peer review of "T-Type Ca2+ Enhancer SAK3 Activates CaMKII and Proteasome Activities in Lewy Body Dementia Mice Model"

_ijms, 2021, doi:10.3390/ijms22126185_

Round 1

Reviewer 1 Report

Xu and coworkers present an interesting study regarding the activation effects of SAK3 over CaMKII and proteasome 2 activities in Lewy body dementia model mice.

The article is nicely written and the conducted research well explained. Moreover the results are interesting. However , the reviewer have some questions and suggestions that could help to improve the manuscript.

1 - It is known the molecular target of SAK3? If yes, it would be interesting to deeply explain it and if not it could be interesting to have at least a computational estimation of the possible binding mode of SAK3. If there are not hypothesis of a concrete target, Some members of the CaMKII-Rpt6 pathway can be tested.

2 -  It is commented that SAK3 improves motor functions. Locomotor impairement is a common problem of many neurodegenrative diseases. Did the authors check the possible effect of SAK3 on other locomotor activity impaired diseases? The authors believes this effect is particular of the studied diseases or can be a broader effect that can be observed in other diseases? 

3 - Somehow related to point2,  it would really interesting if the authors can  elaborate a little bit more this idea "If SAK3 administration promotes the degradation of misfolded proteins, the therapeutics have the potential to solve the problems of diverse protein misfolding diseases such as PD, LBD, tauopathies, and Huntington diseases in addition to AD"

4 - There exist other moleucles with a similar effect to SAK3 at least on some of the analyzed points? It would be nice to have a control to compare the observed SAK3 activity.

5 - Did the authors test the possible effect of SAK3 as mGluR modulator? As authors comment that SAK3 triggers intracellular calcium influx and promotes Glu release, it could be an option.

Author Response

Xu and coworkers present an interesting study regarding the activation effects of SAK3 over CaMKII and proteasome 2 activities in Lewy body dementia model mice.

The article is nicely written and the conducted research well explained. Moreover the results are interesting. However , the reviewer have some questions and suggestions that could help to improve the manuscript.

1 - It is known the molecular target of SAK3? If yes, it would be interesting to deeply explain it and if not it could be interesting to have at least a computational estimation of the possible binding mode of SAK3. If there are not hypothesis of a concrete target, Some members of the CaMKII-Rpt6 pathway can be tested.

Ans: This is important question to solve the mechanism of SAK3. SAK3 enhances the Ca2+ currents in both Cav3.1 and Cav3.2 T-type calcium channels in neuro2A cells over expressed Cav3.1 and Cav3.3 (Yabuki and Fukunaga et al, Neuropharmacology 2017;117:1-13). The application of SAK3 does not induce the significant change in activation and inactivation curves, and promotes Cav3.1 and Cav3.3 currents without affecting channel biophysical properties in the patch crump methods. The activation of CaMKII in the hippocampus was completely blocked the T-type calcium channel blockers in vitro and in vivo animal studies, suggesting that CaMKII-Rpt6 activation is secondary effects through T-type calcium channel enhancer, SAK3. The further studies are required to define the binding site and mode of SKK3 in the T-type calcium channel molecules.

2 -  It is commented that SAK3 improves motor functions. Locomotor impairement is a common problem of many neurodegenrative diseases. Did the authors check the possible effect of SAK3 on other locomotor activity impaired diseases? The authors believes this effect is particular of the studied diseases or can be a broader effect that can be observed in other diseases? 

Ans: We agree with the comment that it can be observed in other diseases. We have been focused on Alzheimer disease and depressive disease because SAK3 improved memory and learning behaviors in those model mice. We believed that other movement disorders such as Huntington’s disease and Parkinson’s disease are improved by degrade the misfolding proteins. We should apply SAK3 in the other protein misfolding diseases in the next steps.

3 - Somehow related to point2,  it would really interesting if the authors can  elaborate a little bit more this idea "If SAK3 administration promotes the degradation of misfolded proteins, the therapeutics have the potential to solve the problems of diverse protein misfolding diseases such as PD, LBD, tauopathies, and Huntington diseases in addition to AD"

Ans: We totally agree with the comment as answer in the above question and will try to the therapeutics for other protein misfolding diseases.

4 - There exist other molecules with a similar effect to SAK3 at least on some of the analyzed points? It would be nice to have a control to compare the observed SAK3 activity.

Ans: This is very important question. SAK3 is the second generated drug to activate T-type calcium channels. The first one is ST101 which is also the enhancer of T-type calcium channel but relatively weak in the T-type calcium channel enhancement in vitro and in vivo studies (Yabuki and Fukunaga et al, Neuropharmacology 2017;117:1-13).

5 - Did the authors test the possible effect of SAK3 as mGluR modulator? As authors comment that SAK3 triggers intracellular calcium influx and promotes Glu release, it could be an option.

Ans: This is again important question. We did the off-target assay (binding assay) using 120 receptors and channels including other calcium channels and glutamate receptor subtypes. There is no specific binding to other molecules except T-type calcium channels. Since SAK3 promotes glutamate release in the hippocampus, we believe SAK3 affects mGlu functions as secondary effects.

Reviewer 2 Report

Review of T-type Ca2+ enhancer SAK3 activates CaMKII and proteasome 2 activities in Lewy body dementia model mice

This manuscript describes the effect of T-type Ca2+ enhancer SAK3 on the formation and degradation of a-synuclein derived Lewy bodies.  The authors extend the work to explore the mechanism of action by examining the effect of SAK3 on the activity of both Ca2+/calmodulin-dependent protein kinase II (CaMKII) and 26S proteosomal activity.  The research was well designed with appropriate controls in place.  The analysis was well done, and the results described in a clear and concise manner.  The text is professionally written with only a few minor changes required:

  • The sentence beginning on line 37 is confusing and should read: “The abnormal metabolism of α-Syn causes the accumulation of misfolded and aggregated α-Syn in LBD and Parkinson's disease (PD)[3].
  • The term αS on line 72 should be α-Syn.
  • The acronyms AMPAR, NMDAR, and nAChR noted in Figure 9 should be defined in the figure legend.

Author Response

Review of T-type Ca2+ enhancer SAK3 activates CaMKII and proteasome 2 activities in Lewy body dementia model mice

This manuscript describes the effect of T-type Ca2+ enhancer SAK3 on the formation and degradation of a-synuclein derived Lewy bodies.  The authors extend the work to explore the mechanism of action by examining the effect of SAK3 on the activity of both Ca2+/calmodulin-dependent protein kinase II (CaMKII) and 26S proteasomal activity.  The research was well designed with appropriate controls in place.  The analysis was well done, and the results described in a clear and concise manner.  The text is professionally written with only a few minor changes required:

  • The sentence beginning on line 37 is confusing and should read: “The abnormal metabolism of α-Syn causes the accumulation of misfolded and aggregated α-Syn in LBD and Parkinson's disease (PD)[3].
  • The term αS on line 72 should be α-Syn.
  • The acronyms AMPAR, NMDAR, and nAChR noted in Figure 9 should be defined in the figure legend.

Ans: Thank you for your encouragement and suggestions. Since the description “The abnormal metabolism of α-Syn causes the accumulation of misfolded and aggregated α-Syn in LBD and Parkinson's disease (PD)[3]” was repeated in the introduction, we omitted the description. In addition, we carefully corrected the test and figure legends according to the comments.